# Post-COVID Endocrine Disorders: Putative Role of Molecular Mimicry and Some Pathomorphological Correlates

**DOI:** 10.3390/diagnostics13030522

**Published:** 2023-01-31

**Authors:** Muslimbek Ghulomovich Normatov, Vadim Evgenievich Karev, Andrey Victorovich Kolobov, Vera Arkad’yevna Mayevskaya, Varvara Aleksandrovna Ryabkova, Vladimir Josefovich Utekhin, Leonid Pavlovich Churilov

**Affiliations:** 1The Laboratory of the Mosaic of Autoimmunity and Department of Pathology, Saint Petersburg State University, 199034 Saint Petersburg, Russia; 2Pediatric Research and Clinical Center for Infectious Diseases, 197022 Saint Petersburg, Russia; 3The Department of English, Saint Petersburg State University of Economics (UNECON), 191023 Saint Petersburg, Russia; 4The Research Institute of Rheumatology and Allergology, First Pavlov State Medical University of Saint Petersburg, 197022 Saint Petersburg, Russia; 5The Department of Pathophysiology with the Course of Immunopathology, Saint Petersburg State Pediatric Medical University, 194100 Saint Petersburg, Russia; 6The St. Petersburg Research Institute of Phthisiopulmonology, 194064 Saint Petersburg, Russia

**Keywords:** antigen mimicry, autoimmune diseases, SARS-CoV-2, human coronaviruses, thyroid gland, pancreatic islets, adrenals, pituitary

## Abstract

In order to identify corresponding amino acid sequences (pentapeptides) between the SPs, MPs and NPs of human coronaviruses and human autoantigens targeted in autoimmune endocrinopathies, and for a comparative analysis of the various coronaviruses proteome and the proteome of human, the original computer program was used. Quantitatively, SP, MP and NP of the human coronaviruses were found to share totally 117 minimal immune pentapeptide epitopes: 79 in SP, 14 in MP and 24 in NP, – with 18 autoantigens expressed by human endocrinocytes. The shared pentapeptides belong to the proteins of human endocrine cells. Samples of the pituitary, adrenal and thyroid from patients who died from coronavirus infection (COVID-19) were studied morphologically using histochemical methods. A high incidence of SARS-CoV-2 infection of endocrine cells was showed. The high affinity of SARS-CoV-2 the cells of the adenohypophysis was revealed, but there was no expression of viral proteins by the cells of the neurohypophysis. The foci of lesions in endocrine organs contained abundant lymphocytic infiltrates which may indicate the impact of autoimmune processes. Autoimmune disorders have a multi-faceted etiology and depend on polygenic predispose and additive action of many epigenetic and environmental factors causing hyperstimulation of imperfectly functioning immune system. It means that the phenomenon of molecular mimicry cannot be blamed as their single prerequisite, but it is just a tile in mosaic of autoimmunity. The facts revealed emphasize the need of endocrinological diagnostic alertness of a physician while observing patients with post-vaccination and post-COVID-19 health disorders.

## 1. Introduction

The novel coronavirus disease pandemic, although currently on the decline, has given rise to a growing problem, namely, post-COVID subacute and chronic health disorders, in particular, various variants of post-COVID syndrome. This syndrome is most frequently observed among those who experienced an acute infection in a non-severe and even mild form, and the number of such patients in the world is increasing [1]. Based on the recent literature, the post-COVID cohort is further divided into two categories: (1) subacute or ongoing symptomatic COVID-19, which includes symptoms and abnormalities present from 4 to 12 weeks beyond acute COVID-19; and (2) chronic or post-COVID-19 syndrome, which includes symptoms and abnormalities persisting or present beyond 12 weeks of the acute COVID-19 onset and are not attributable to alternative diagnoses [2]. A recent meta-analysis revealed that 80% of the patients infected with SARS-CoV-2 developed one or more long-term symptoms [3]. Neuropsychiatric, rheumatic, respiratory, cardiovascular, gastrointestinal and, not infrequently, endocrine manifestations were identified, and over 40% of patients still reported fatigue during 7–9 months after the acute COVID-19 resolution, and the prevalence of most symptoms has even risen after 7–9 months compared to the early recovery period (3–10 weeks) [4]. Pathological autoimmunity plays a pivotal mechanistic part in post-COVID complications [5]. Another growing cohort includes individuals with post-anti-COVID vaccination adverse manifestations which presumably have an autoimmune/autoinflammatory origin [6]. Quite often, the post-COVID syndrome, as well as anti-COVID vaccination adverse effects, alter the endocrine regulation, including thyroid [7], pituitary [8], adrenal [9], insulin dependent [10] and gonadal [11] mechanisms.

The molecular mimicry between pathogen and host antigens has been long time suspected as a provocative mechanism of pathological autoimmunity triggered by infection [12,13]. That assumption is true for the COVID-19 sequela as well [14]. Previously, we have demonstrated peptide sharing between immunogenic epitopes of *SARS-CoV-2 spike protein (SP)* and few autoantigens of human endocrine cells [15,16]. In addition to molecular mimicry, among the concepts of autoimmunology, other options are considered in the literature as specific mechanisms for the etiology of autoimmunity during the induction of autoimmunity by viruses: epitope spreading [17], cryptic epitopes [18] and bystander activation [19]. Each of these mechanisms is regarded as a likely etiological mechanism responsible for the activation of self-reactive (Self-Reactive) immune cells. Moreover, a combination of the options mentioned above should not be excluded.

Along with the SARS-CoV-2 SP, several other viral antigens have been proven to be highly significant in the immune response against COVID-19. The *SARS-CoV-2 Nucleocapsid protein (NP)* is its immunodominant antigen; moreover, anti-NP antibodies have the highest titers among all anti-viral specificities in persons previously infected with this virus [20]. The *SARS-CoV-2 Membrane protein (MP)* is essential in the disease pathogenesis due to its role in anti-interferonogenic and pro-apoptotic effects and also is highly immunogenic [21,22]. In this study, we have explored NP and MP for antigen mimicry with several common target antigens of human autoimmune endocrinopathies (hypophysitis, adrenalitis, insulitis and oophoritis/orchitis).

The autoimmune complications were suspected after coronavirus infections even prior to current pandemic of SARS-CoV-2. For example, in 2004, molecular mimicry was hypothesized between SARS-CoV-1 and pituitary antigens [23]. Later, seasonal coronaviruses were blamed for possible provocation of CNS lesions *via* autoimmunity [24]. That is why we also compared the molecular mimicry potential of SPs from all other known human coronaviruses against typical autoantigens of human endocrinocytes.

In this article, we have focused on the putative role of molecular mimicry and some pathomorphological correlates. 

## 2. Materials and Methods

In order to identify matching amino acid sequences (pentapeptides) between the SPs, MPs and NPs of human coronaviruses and human autoantigens targeted in autoimmune endocrinopathies, and for a comparative analysis of the various coronaviruses proteome and the proteome of human, we used the original computer program ALIGNMENTAJ (https://github.com/muslimb/MyProekt1 (accessed on 16 October 2022)) created by M.G.Normatov. The ALIGNMENTAJ (accessed on 16 October 2022) works fast and uses an algorithm with local alignments. The data from the UniProt (https://www.uniprot.org/ (accessed on 12 December 2003)) database about primary structure of proteins were used.

A comparison of proteins by a consecutive search for regions of one protein in the others, which is essentially a standard task of finding a sub-string in a string, was performed. This algorithm is implemented in standard methods of Python (https://www.python.org/ (accessed on 20 February 1991)), in which the main program was coded. A bioinformatics analysis of the probable pentapeptide sharing between human endocrinocytes’ antigens and various proteins of coronaviruses was based on the NCBI (https://www.ncbi.nlm.nih.gov/ accessed on 4 November 1988), UniProt (accessed on December 2003) and Immune Epitope Data Base (https://www.iedb.org/home_v3.php (accessed 14 January 2008)) databases. Matching peptides of human endocrinocytes and SP, MP and NP of various coronaviruses were analyzed using pentapeptides as sequence probes since a peptide grouping formed by the five amino acid (aa) residues defines a minimal immune determinant that can (1) induce highly specific Abs, and (2) determine antigen-Ab specific interaction [13]. A library of human endocrinocyte-associated proteins has been assembled at random from the UniProtKB database [25]. The primary sequence of all viral proteins was dissected into pentapeptides offset by one residue and the resulting viral pentapeptides were analyzed for their occurrences within a primary sequence of human autoantigens explored. Occurrences and corresponding proteins were annotated. The immunological potential of the pentapeptides shared between various coronaviruses and human endocrinocytes was analyzed by searching the Immune Epitope Data Base (IEDB) [26] for the immunoreactive epitopes of various coronaviruses hosting the shared pentapeptides. 

In order to search for pathomorphological correlates of bioinformatics findings, a histological analysis of some endocrine organs was carried out. For morphological studies samples of the pituitary, adrenal and thyroid glands taken from deceased adults who died from a new coronavirus infection (COVID-19), confirmed by intravital and post-mortem isolation of SARS-CoV-2 nucleotide sequences from biological material by PCR. In all described cases, COVID-19 infection acted as an immediate cause of death confirmed by authorized protocol of autopsy. Post-mortem examination revealed the absence of other clinically significant pathology. The mechanism of death in all cases was associated with respiratory failure and multi-organ damage due to hypoxia. Supravital radiological methods (CT) as well as autopsy investigation confirmed the fact that endocrine organs were involved in viral inflammation.

The c samples were fixed with a 10% buffered formalin solution (pH = 7.4) within 72 h. After fixation, the tissue samples were dehydrated using isopropanol, paraffin imbibition, and embedded in paraffin according to conventional methods. Tissue sections 4 μm thick were made from paraffin blocks and placed on glass slides treated with polylysine. Immunohistochemical studies were performed using the A360 Immunohistosteiner (Thermo, Waltham, Massachusetts, USA), the UltraVision Quanto DAB universal immunohistochemical imaging system (Thermo, USA), SARS-CoV-2 Spike rabbit polyclonal antibodies (GeneTex, Irvine, California, USA), and mouse monoclonal antibodies to Caspasae3 for the purpose of identifying apoptotic mechanism of cell death (clone 3CSP03, Diagnostic BioSystems, USA) in accordance with the recommendations of the reagent manufacturers.

## 3. Results

### 3.1. Bioinformatics Analysis

Quantitatively, SP, MP and NP of the human coronaviruses were found to share totally 117 minimal immune pentapeptide epitopes: 79 in SP, 14 in MP and 24 in NP, – with 18 autoantigens expressed by human endocrinocytes. The shared pentapeptides belong to the proteins of human endocrine cells listed in Table 1, Table 2 and Table 3.

The Immune Epitope Data Base exploration [26] revealed that all of the shared pentapeptides described in Table 1, Table 2 and Table 3 belong to those epitopes of SP, MP or NP from various human coronaviruses that have been experimentally validated as immunoreactive ones.

We have shown the location of pentapeptides in 3D structures of spike proteins in Figure 1 using PDB [27] and AlphaFold [28] databases.

### 3.2. Pathohistological Data

Immunohistochemical study of the organs of the endocrine system (pituitary, thyroid and adrenal glands) taken on autopsies from the patients who had a severe lethal course of COVID-19 showed a high incidence of SARS-CoV-2 infection of their endocrine cells (Figure 2). There was the abundant expression of viral proteins by the cells of the adenohypophysis and complete absence of its expression by the cells of the neurohypophysis (Figure 2A–D).

The absence of Caspasae3 expression by degenerative and dying endocrinocytes was revealed (Figure 2E–H).

The manifestations of virus-induced transformation of parenchymal cells in endocrine organs infected with SARS-CoV-2 appeared sharply. Obvious infiltration of the stroma and parenchyma of the endocrine organs with lymphocytes was observed (Figure 3).

## 4. Discussion

Although pathogenesis of COVID-19 sequela is still far from entire comprehension, it is generally accepted that antiviral protection is activated as a result of viral PARPs recognition by various variants of body pattern-recognizing receptors (PRRs): Toll-like receptors (TLRs), RIG-I-like receptors (RLR) and the cytoplasmic proteins family receptor involved in caspase activation (NALP), in ensemble causing involvement of both paleo- and neo- immunity components and resulting in the development of inflammation [29,30,31].

The overcoming of antiviral defense by viruses is associated with the evolutionary developed strategies of pathogens to influence the immune system so that allows the virus to ensure its sufficiently effective reproduction. To understand the mechanisms for achieving this goal, searches are underway for the mimicking amino acid sequences in viral proteins and in human immune system proteins suggesting that it is precisely homologous amino acid sequences in viral proteins that can unbalance immune regulatory mechanisms and cause a wide variety of immunopathological consequences: from immunosuppression to a cytokine storm.

Among the ideas about the etiology of autoimmunity caused by viruses (epitope spreading, cryptic epitope, bystander activation, molecular mimicry), the concept of molecular mimicry attracts no less attention than other concepts. 

Currently, there are four major criteria for identifying molecular mimicry as a provoking mechanism of autoimmune diseases [32]: (1) “similarity between the host epitope and the epitope of a microorganism or environmental agent”, (2) “detection of antibodies or T-cells that cross-react with both epitopes in patients with autoimmune disease”, (3) “epidemiological link between the exposure to an environmental agent or microbe and the development of autoimmune disease”, and (4) “reproducibility of autoimmunity in an animal model following sensitization with appropriate epitopes either after infection with a microbe or exposure to an environmental agent”. To date, to our knowledge, only the first and third criteria are satisfied for the post-COVID syndrome or COVID-19 vaccine-associated side effects.

At the same time, there are some concerns regarding the molecular mimicry concept itself. First of all, humans are challenged with multiple infections throughout their lives, including infections with the pathogens whose antigens cross-react with the human ones, but not all infected humans develop autoimmune diseases. For example, Kanduc et al. [33] demonstrates that pentapeptides of 30 common viruses are disseminated throughout practically the entire human proteome, and each viral pentapeptide is repeated almost more than 10 times. This massive viral-human peptide overlapping calls into doubt the possibility of the direct causal association between the virus–host sharing of amino acid sequences and the incitement to autoimmune reactions. Indeed, autoimmune diseases should theoretically approach a 100% real incidence according to Kanduc et al. [33], since the 30 viruses they examined practically are more or less disseminated throughout the entire mankind. Moreover, two years later, Trost et al. [34] examined 40 pathogenic and nonpathogenic bacterial proteomes for the amino acid sequence similarity to human proteome and reported that one third of human proteins shares at least one nonapeptide with someone of these bacteria. Then the authors detailed the bacterial peptide overlapping with human proteome at the penta-, hexa-, hepta- and octapeptide levels using exact peptide matching analysis and demonstrated that virtually every human protein contains a bacterial pentapeptide or hexapeptide motif [35].

Besides that, although for a long time it was believed that T-cell receptors recognize sequential determinants only, several recent lines of evidence have demonstrated that the T-cell cross-reactivity analyses could not rely on sequence similarity alone [36]. It was shown that individual T-cell receptors could recognize different peptide/MHC complexes that do not show strong sequence homology, and it was suggested that structural criteria rather than primary sequence might be critical for the T-cell receptor recognition.

In spite of all the above-mentioned doubts, the fact of molecular mimicry displayed by the immunoreactive epitopes of SARS-CoV-2 proteins with marker autoantigens of common human endocrinopathies maybe of some prognostic significance. Indeed, 17-alpha-hydroxylase is an autoimmunity target in diseases involving steroid-producing cells, especially in Addison’s disease, autoimmune polyglandular syndrome type I, and premature hypogonadism [37,38,39]. Carboxypeptidase E is a target of autoimmunity in late-onset (latent) autoimmune diabetes of adults with some diagnostic value to distinguish it from diabetes mellitus type 2 [40]. The same is true for receptor-type protein-tyrosine-phosphatases which are targeted in autoimmune diabetes mellitus [41]. Finally, the cytotoxic T-lymphocyte 4 antigen (CTLA-4) is expressed in pituitary, and antibodies against this target induce autoimmune hypophysitis [42]. The involvement of adenohypophysis and adrenals in antigen mimicry and in immunopathological inflammation during severe COVID-19, which we have demonstrated, seems to be very important for untoward lethal course of disease, because it apparently may jeopardize the appropriate defensive stress-related mechanisms protecting from systemic hypercytokinemia and related hemodynamic shock [15]. In our studies, the most pathogenic among human coronaviruses (SARS-CoV and MERS-CoV) possessed the greatest number of epitopes shared with human endocrinocytes. SP of various coronaviruses shared greatest number of epitopes with endocrine autoantigens, compared to their MP or NP. Thyroid autoantigens were most active in “sharing” their epitopes with highly pathogenic coronaviruses, which corresponds to growing number of cases of autoimmune thyroid diseases cases described during and after COVID-19 [43] and even few of them, after anti-COVID-19 vaccination [44]. At the same time, seasonal coronaviruses quite often shared their immunodominant pentapeptides with several autoantigens of pancreatic islet β-cells. Before the current pandemic, they rarely attracted attention as potential viral diabetogens, although in veterinary medicine, insulin dependent diabetes due to seasonal coronavirus was described in a foal [45]. Interestingly, neurohypophysis was free from any histological signs of involvement in severe COVID-19, as well as its autoantigen, rabfillin-3A, which displayed only one pentapeptide shared and with one seasonal human coronavirus only. These data may be related to the recently discovered protective role in COVID-19 played by oxytocin secreted by neurohypophysis [46].

Finally, we consider bioinformatics analysis to be an essential step in the preliminary evaluation of the risks and autoimmunity spectrum in COVID-19 complications, including the post-COVID-19 syndrome. A similar point of view recently was emphasized by other researchers [47]. Additionally, it may be useful in epitope selection for elaboration of the safest anti-COVID-19 vaccines S [48]. 

Immunohistochemical study of the endocrine system (autopsied pituitary, thyroid and adrenal glands) from the patients who had a severe lethal course of COVID-19, showed high tissue concentration of SARS-CoV-2 infection. However, the affinity of SARS-CoV-2 to endocrine cells was selective. Thus, the abundant expression of viral proteins by the adenohypophysis cells was revealed. Conversely, there was no expression of viral proteins by the cells of the neurohypophysis. It is interesting that product of neurohypophysis oxytocin is successfully used in clinical treatment of COVID-19 [49].

The obtained results may be explained due to the possible absence in the neurohypophysis cell membrane such proteins as APC 2, NRP 1 or serine protease TMPRSS2, that are presented in the type 2 pneumocytes, enterocytes of the small intestine, endotheliocytes, smooth myocytes, neurons of the cerebral cortex, striatum, hypothalamus, brain stem. These proteins are crucial for the penetration of SARS-CoV-2 into the cell.

SARS-CoV-2 infection of endocrine cells did not lead to their apoptotic death, as shown by the absence of Caspasae3 expression. It can possibly be explained due to other mechanisms of death or other forms of apoptosis prevailing in cells affected by COVID-19. It should be taken into account that the absence of Caspase-3 expression does not entirely exclude the possibility of apoptotic mechanisms of cell death, because in some pathways of apoptosis, the process can bypass the Caspase-3 link [50]. At the same time, virus-induced transformation of parenchymal cells in endocrine organs infected with SARS-CoV-2 was described. Lymphocyte infiltration of the stroma and parenchyma of the endocrine organs was observed, which usually is associated with a cell-mediated immune mechanisms of their lesion.

## 5. Conclusions

Autoimmune disorders have a multi-faceted etiology and depend on polygenic predispose and additive action of many epigenetic and environmental factors causing hyperstimulation of imperfectly functioning immune system. It means that the phenomenon of molecular mimicry cannot be blamed as their single prerequisite, but it is just a tile in mosaic of autoimmunity.

## Figures and Tables

**Figure 1 diagnostics-13-00522-f001:**
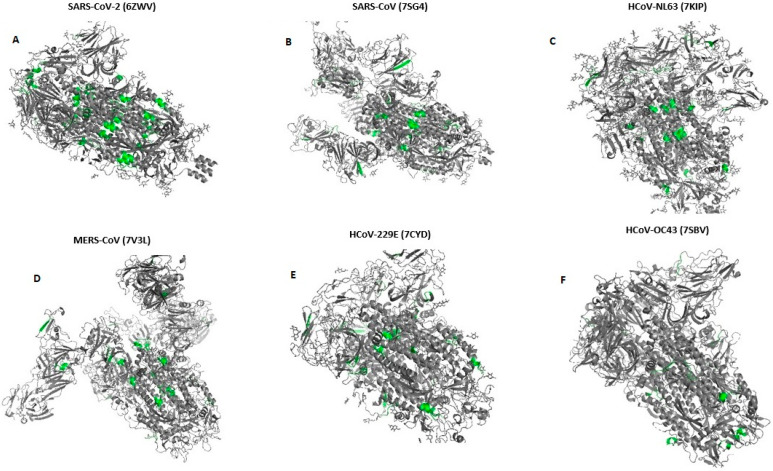
Location of pentapeptides shared with human endocrine autoantigens in 3-D structure spike glycoproteins of all Human Coronaviruses (**A**–**F**). (According to databases: PDB [27] and AlphaFold [28]). Pentapeptides are shown with grass-green.

**Figure 2 diagnostics-13-00522-f002:**
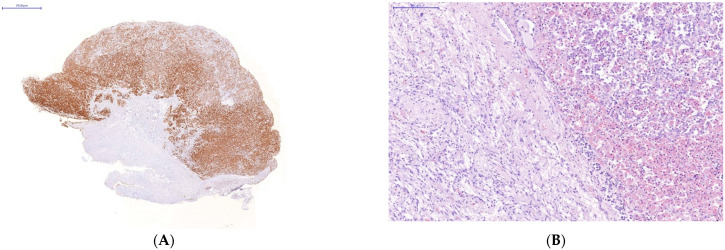
Changes in some endocrine organs of patients who died from COVID-19 infection. Uneven expression of SARS-CoV-2 spike antigen by pituitary cells (**A**,**B**): abundant expression by adenohypophysis cells (**C**) and no expression by neurohypophysis cells (**D**); focal expression of the SARS-Cov2 spike antigen by a group of degeneratively altered *Caspasae3-negative* adrenal parenchymal cells, arrows (**E**,**F**); expression of SARS-CoV-2 spike antigen by *Caspasae3-negative* thyroid follicle cells (**G**,**H**). (**A**,**C**–**E**,**G**)—immunohistochemistry (IHH), rabbit polyclonal anti-SARS-CoV-2 Spike (GeneTex, USA), DAB; (**F**,**H**)—IHH, mouse monoclonal anti-Caspasae3 (clone 3CSP03, Diagnostic BioSystems, USA); (**B**)—H&E. The length of the scale segment (**A**)—2000 μm, (**B**–**D**)—200 μm; (**E**,**F**–**H**)—100 μm.

**Figure 3 diagnostics-13-00522-f003:**
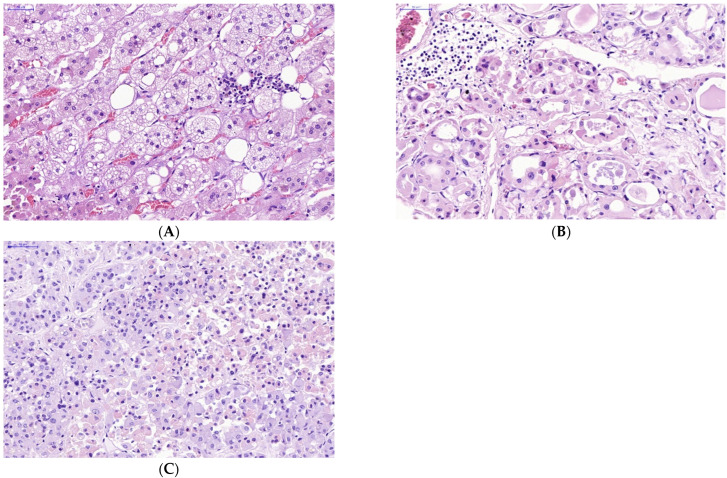
Lymphocyte infiltration in stroma and parenchyma of some endocrine organs in victims of lethal COVID-19. (**A**—adrenal gland, **B**—thyroid gland, **C**—pituitary gland) (**A**–**C**)—H&E. Scale bar length: (**A**–**C**)—50 μm).

**Table 1 diagnostics-13-00522-t001:** Spike glycoprotein of human coronaviruses and autoantigens common for autoimmune endocrinopathies: shared pentapeptides. ID numbers of immunoreactive epitopes from the Immune Epitope Data Base for Human Coronaviruses spike proteins are shown in the square brackets for each pentapeptides.

Spike glycoprotein							
Hum. Coronaviruses//Autoantigens of Endocrine Cells	SARS-CoV-2 (P0DTC2)	SARS-CoV (P59594)	MERS-CoV (R9UQ53)	HCoV-HKU-1 (S5YA28)	HCoV-OC43 (P36334)	HCoV-NL63 (Q6Q1S2)	HCoV-229E (P15423)
Thyroid peroxidase (P07202)	RAAEI [39003]	RAAEI [51379]	-	-	-	SFSKL [1662988]	-
Thyrotropin receptor (P16473)	LLPLV [1071428], ICGDS [1312282]	ICGDS [1418339]	DTKIA [1411039], ASELS [1411879]	-	-	ILLVL [-]	VSQTS [1526382], IPSLP [1510901]
Thyroglobulin (P01266)	LDSKT [1087553], FNFSQ [1310327], SAIGK [1310542]	FNFSQ [1087693], FLLFL [41504]	GFGGD [1451681]	WYQKP [1510167]	VVSCL [1419202], RVSPG [1646031]	LQENQ [1410576], LKSGV [1411387]	FVNTT [1415918], LQENQ [1471298]
Alpha-enolase (P06733)	-	-	-	-	-	IADLA [1418283]	IADLA [1385058]
RPH3L (**rabfillin-3a**) (Q9UNE2)	-	-	RLDVL [1439102], LDVLE [1455525], DVLEQ [1439102]	-	-	-	-
Cytotoxic T-lymphocyte protein 4 (P16410)	-	-	-	-	-	-	-
Prolactin (P01236)	SNLLL [1309139]	-	-	-	-	TEVRG [1513648], NLSSE [1473904]	-
Steroid 21-hydroxylase (P08686)	LQDVV [1310833]	LQDVV [14208]	GTVII [1406455]	-	PDLSL [1411536]	-	-
Steroid 17-alpha-hydroxylase (P05093)	-	-	-	DTLMQ [1424576]	-	-	EISTL [1427980]
Glutamate decarboxylase 1 (Q99259)	VGYQP [1310441], AGAAL [23293]	AGAAL [62872]	-	-	-	-	DGDGI [1475433]
Glutamate decarboxylase 2 (Q05329)	-	-	-	-	-	-	-
Receptor-type tyrosine-protein phosphatase-like N (Q16849)	LPPLL [3982]	LPPLL [3983]	PLGQS [1419664], LVALA [-]	LSTLL [1507146], GSSSR [1444184]	LPPLL [1441061], EPALL [1408158], LAGVA [-]	RLAAL [-]	RLAAL [36723]
Receptor-type tyrosine-protein phosphatase N2 (Q92932)	-	-	AALSA [1503295]	LMQGV [1424576], SSSRS [1444185], GAALA [1432711]	-	-	AVLTY [1648957]
Islet cell autoantigen 1 (Q05084)	LDPLS [1310302], GYQPY [1074918]	GYQPY [19657], KGYQP [16417], ELLNA [23437]	FRKVQ [1519626]	NASSL [1501401]	ASLQE [1412712]	-	-
Insulin (P01308)	-	-	-	-	-	-	-
Insulin receptor (P06213)	-	-	DYYRK [1414267], LKELG [1422379]	FRDLS [1462061], TICKS [1448456], RKRRS [1489830]	LKDGV [1459900], ENNVV [1455023]	IVNLL [1509536], SNSSS [1418007]	-
Zinc transporter 8 (Q8IWU4)	-	-	LLSLFS [1482114]	VSSCA [1423081], ALLSI [1646363]	-	-	-
Carboxypeptidase E (P16870)	SALLA [1069137]	-	-	-	-	-	-

**Table 2 diagnostics-13-00522-t002:** Membrane proteins of human coronaviruses and autoantigens common for autoimmune endocrinopathies: shared pentapeptides. ID numbers of immunoreactive epitopes from the Immune Epitope Data Base for Human Coronaviruses membrane proteins are shown in the square brackets for each pentapeptides.

Membrane protein							
Hum. Coronaviruses//Autoantigens of Endocrine Cells	SARS-CoV-2 (P0DTC5)	SARS-CoV (P59596)	MERS-CoV (R9UNX5)	HCoV-HKU-1 (Q0ZJ82)	HCoV-OC43 (Q01455)	HCoV-NL63 (U3M6Q8)	HCoV-229E (P15422)
Thyroid peroxidase (P07202)	-	-	-	-	-	-	-
Thyrotropin receptor (P16473)	-	-	TSVTA [1488579]	-	-	-	-
Thyroglobulin (P01266)	-	-	-	-	-	-	LFRRA [-]
Alpha-enolase (P06733)	-	-	-	-	-	-	-
RPH3L (**rabfillin-3a**) (Q9UNE2)	-	-	-	-	RLPST [1498820]	-	-
Cytotoxic T-lymphocyte protein 4 (P16410)	-	-	-	-	-	-	-
Prolactin (P01236)	-	-	-	-	-	-	-
Steroid 21-hydroxylase (P08686)	-	-	-	-		-	-
Steroid 17-alpha-hydroxylase (P05093)	-	-	-	-	-	-	-
Glutamate decarboxylase 1 (Q99259)	-	-	-	-	-	-	DGDGI [-]
Glutamate decarboxylase 2 (Q05329)	-	-	-	LWLMW [-]	LWLMW [-]	-	-
Receptor-type tyrosine-protein phosphatase-like N (Q16849)	TRPLL [1312642]	-	PLVED [1443619]	-	-	-	EVNAI [1414330]
Receptor-type tyrosine-protein phosphatase N2 (Q92932)	-	-	-	-	-	-	EVNAI [1414330]
Islet cell autoantigen 1 (Q05084)	-	-	-	-	-	-	-
Insulin (P01308)	-	-	-	-	-	-	-
Insulin receptor (P06213)	-	-	-	TVIRG [1646603]	-	-	TVAVP [1470736]
Zinc transporter 8 (Q8IWU4)	-	-	-	-	-	-	-
Carboxypeptidase E (P16870)	GNYKL [56634]	GNYKL [1531436]	-	-	-	-	-

**Table 3 diagnostics-13-00522-t003:** Nucleoproteins of human coronaviruses and autoantigens common for autoimmune endocrinopathies: shared pentapeptides. ID numbers of immunoreactive epitopes from the Immune Epitope Data Base for Human Coronaviruses nucleoproteins are shown in the square brackets for each pentapeptides.

Nucleoprotein							
Hum. Coronaviruses//Autoantigens of Endocrine Cells	SARS-CoV-2 (P0DTC9)	SARS-CoV (P59595)	MERS-CoV (T2BBK0)	HCoV-HKU-1 (S5ZBQ7)	HCoV-OC43 (P33469)	HCoV-NL63 (H9EJA4)	HCoV-229E (P15130)
Thyroid peroxidase (P07202)	-	-	-	-	PTSGV [1411373]	-	AAALK [1406449]
Thyrotropin receptor (P16473)	-	-	-	-	ILNKP [1430757], DVYEL [1429244]	RGQRV [1429742]	-
Thyroglobulin (P01266)	-	RVRGG [51074]	-	SASNS [1501942], FTVST [1437039]	-	-	-
Alpha-enolase (P06733)	-	-	-	-	GKDAT [1409816]	-	-
RPH3L (**rabfillin-3a**) (Q9UNE2)	-	-	-	-	-	-	-
Cytotoxic T-lymphocyte protein 4 (P16410)	PPTEP [25542]	PPTEP [33669]	-	-	-	-	-
Prolactin (P01236)	-	-	-	-	-	-	-
Steroid 21-hydroxylase (P08686)	-	-	-	-	-	-	-
Steroid 17-alpha-hydroxylase (P05093)	ALLLL [2431]	ALLLL [19442]	-	-	-	-	-
Glutamate decarboxylase 1 (Q99259)	-	DNVIL [7807]	-	-	-	-	-
Glutamate decarboxylase 2 (Q05329)	-	-	-	-	-	-	-
Receptor-type tyrosine-protein phosphatase-like N (Q16849)	-	-	SSSRA [1440437], SEPPK [1423544]	-	SSSRA [1436882]	-	-
Receptor-type tyrosine-protein phosphatase N2 (Q92932)	SSSRS [60669]	SSSRS [4782]	-	-	-	REENV [1490443]	RAPSR [1407387]
Islet cell autoantigen 1 (Q05084)	-	-	-	-	-	-	LGKFL [1421399]
Insulin (P01308)	-	-	-	-	-	-	-
Insulin receptor (P06213)	-	-	-	-	-	-	-
Zinc transporter 8 (Q8IWU4)	-	-	-	-	-	-	-
Carboxypeptidase E (P16870)	SSPDD [1075010]	-	DLQGN [1422553]	-	-	-	-

## Data Availability

Not applicable.

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
