# Peer review of "Post-COVID Endocrine Disorders: Putative Role of Molecular Mimicry and Some Pathomorphological Correlates"

_diagnostics, 2023, doi:10.3390/diagnostics13030522_

Round 1

Reviewer 1 Report

The authors of manuscript, used bioinformatics softwares to study the relation of autoimmune disease cause by SARS-CoV-2 infection. The authors did bioinformatic analysis of pentapeptide sharing between human autoantigens of endocrinocytes and SARS-CoV-2 NP, S and M protein. The results of their study shows total of six human-identical were found in SARS-CoV-2 membrane and Nucleocapsid proteins. All shared epitopes belong to antigens of endocrine commonly targeted autoimmune endocrinopathies. The authors also collected adrenal and thyroids samples from a dead patient from covid-19 to look at the effect of SARS-CoV-2 infection on these organs using immunohistology. 

The authors used a proper methodology to design the study and results written clearly. The authors discussed the finding in details with supporting statements. Overall the manuscript written well and I have only one suggestion for the authors to improve their manuscript. On discussion part the authors mostly focus talking about their bioinformatic data, it would be great if they correlate their bioinformatics finding with immuohistology data.

Author Response

We are extremely grateful to the Reviewer for the high assessment of our article and the deep and favorable analysis.

Point 1: The authors of manuscript, used bioinformatics softwares to study the relation of autoimmune disease cause by SARS-CoV-2 infection. The authors did bioinformatic analysis of pentapeptide sharing between human autoantigens of endocrinocytes and SARS-CoV-2 NP, S and M protein. The results of their study shows total of six human-identical were found in SARS-CoV-2 membrane and Nucleocapsid proteins. All shared epitopes belong to antigens of endocrine commonly targeted autoimmune endocrinopathies. The authors also collected adrenal and thyroids samples from a dead patient from covid-19 to look at the effect of SARS-CoV-2 infection on these organs using immunohistology. 

The authors used a proper methodology to design the study and results written clearly. The authors discussed the finding in details with supporting statements. Overall the manuscript written well and I have only one suggestion for the authors to improve their manuscript. On discussion part the authors mostly focus talking about their bioinformatic data, it would be great if they correlate their bioinformatics finding with immuohistology data.

 Response 1: Thanks for advice. We Have Accomplished this.

Reviewer 2 Report

I read with interest the manuscript entitled Post-COVID Endocrine Disorders: Putative Role of Molecular Mimicry and Some Pathomorphological Correlates. The authors present a very interesting topic to discuss and show evidence of similarities in pentapeptides between cells from different endocrine organs and coronaviruses. Additionally, authors present evidence of infection in these same organs from cadavers that suffered from a severe COVID-19, lymphocyte infiltration, and absence of the expression of caspase-3. 

Here are specific comments by section:

ABSTRACT

1.     There is no clarity about the present work, there is mixed information about previous and current work from the same group of authors.

INTRODUCTION

1.     There is mention of molecular mimicry as mechanism associated to the development of autoimmunity after infection, but they do not mention other mechanisms such as bystander activation.

2.     Authors claim that there is a provocative mechanism of pathological autoimmunity triggered by infection with COVID-19, however this is not sustained by the references 14-15, rather reference number 14, reported on peptide sharing between SARS-CoV-2 spike glycoprotein and surfactant-related proteins. And reference 15, reported on 7–9 residue matches in several known human proteins with a 15mer palindromic SARS-CoV-2 peptide. But neither of these references, proves the induction of autoimmunity by the presence of the infectious agent.

METHODOLOGY:

1.     Authors used ALIGNMENTAJ, created by M.G.Normatov. However, there is no information about the performance of this tool in other studies. I tried to search for additional articles using this tool and was not able to find any. If authors could present some information about this, it would be reassuring.

2.     Immunohistochemical analysis searched for SARS-CoV-2 Spike protein and Caspase 3. However, authors stated that the aim of this manuscript was to explore NP and MP for antigen mimicry with several common target antigens of human autoimmune endocrinopathies (hypophysitis, adrenalitis, insulitis and oophoritis/orchitis). Why are authors including information about the SP, when it was not considered in the objective? Is it because this information has been partially published before (refence #17)?

3.     The Immunohistochemical analysis was performed in endocrine organs of patients that died from COVID, although they only confirmed SARS-CoV-2 positivity but no cause of death, patients could have died from different causes and be positive to COVID-19. This is only stated in the results section but should be specified in the methodology section. There is no information about the presence of clinical signs of autoimmunity in these patients either.

4.     The bioinformatic analysis as well as table 5, contain information about the spike protein which have already been published by the same group of authors (reference #17 of the manuscript). This information should be removed as is not in accordance with the objective of the present paper and has already been published.

5.     What is the relevance of table 4, 5 and 6. This information is already available in previous tables. The ID in IEDB could just be added to each sequence in previous tables.

6.     Authors state that the absence in the expression of caspase 3 by degenerative and dying endocrinocytes (Fig. 2, I-H), is most likely due to other mechanisms of death, and not apoptosis, but other forms of apoptosis have been demonstrated by decreased levels of Bcl-2 and increased Bax, that directly activates caspase 7 and 9. (Oncogene 20, 6570-6578 (2001) https://doi.org/10.1038/sj.onc.1204815

7.     Figure 2. Letters A, B, and C, D, seem to present the same information. Letters I and F are supposed to have arrows; however, they are not visible. Letters I-H refer to the expression of the spike protein, not included in the objective of the present paper.

DISCUSSION

1.     There is still and ongoing discussion about the origins of autoimmunity, and although some authors support molecular mimicry, there are several studies that oppose this mechanism as the origin of autoimmunity and favor others such as bystander activation, or even a combination of both. 

2.     Authors from this paper consider molecular mimicry as the sole explanation for the development of autoimmunity but this is far from being generally accepted, mainly because there is still a missing link since more than often mimicry is not enough to ensure an autoimmune response. Apart from rheumatic fever and a few other cases, this mechanism has not been proven to be the origin of other autoimmune diseases, and even less in the case of systemic autoimmune diseases, in experimental models there is always the need to induce an artificial severe inflammatory response for autoimmunity to develop, the single administration of the “autoantigen” almost never is considered enough for the animal to develop autoimmunity, and this should be considered by the authors, otherwise the information contained in the paper might be misleading. 

3.     Authors also claim than an unbalance in the regulatory mechanisms of the immune response triggered by the mimicking can be responsible for immunosuppression to the development of a cytokine storm, nevertheless this is too broad and not just the regulatory mechanisms have been implicated in autoimmunity, but other parts of the immune response including the innate and the adaptive have also been linked to the development of autoimmunity. 

4.     What seems to be interesting though is that in SARS-CoV-2 infection particularly, is that the infection frequently triggers a very important inflammatory response, and this could certainly lead to a bystander activation, which in conjunction to molecular mimicry could lead to an autoimmune response, this could be added to the discussion.

5.      The current criterion for identifying molecular mimicry is from a book published in 2007 (reference #29), to this respect, only the first has been really proved, since the epidemiological link is still debatable.

6.     Although authors mention the importance of structural criteria rather than primary sequence for TCR recognition, there is no account for this in the results of the paper, and this is of upmost importance, since peptides might be hidden or even susceptible to suffer from PTMs and their recognition is not always achieved. 

7.     The presence of the virus and following infiltration of cells might be just a reflection of infection, not really a reflection of autoimmunity, as is being stated by the authors. Therefore, authors must support why they choose the later and not the first as a possible explanation for this finding.

8.     Authors mention a few “marker autoantigens” associated with autoimmunity but not all where shared with SARS-CoV-2 or any other coronavirus (17-alpha-hydroxylase and CTLA-4).

9.     The lack of expression of proteins from SARS-CoV-2 in the neurohypophysis could be explained by a lack of receptors ACE2 or NRP-1? And if so, what would be the clinical relevance of such distinction in expression?

CONCLUSION

1.     The paper lacks a conclusion.

Author Response

Response to Reviewer 2 Comments

We are extremely grateful to the Reviewer 2 for the high assessment of our article and the deep and favorable analysis.

I read with interest the manuscript entitled Post-COVID Endocrine Disorders: Putative Role of Molecular Mimicry and Some Pathomorphological Correlates. The authors present a very interesting topic to discuss and show evidence of similarities in pentapeptides between cells from different endocrine organs and coronaviruses. Additionally, authors present evidence of infection in these same organs from cadavers that suffered from a severe COVID-19, lymphocyte infiltration, and absence of the expression of caspase-3. 

Here are specific comments by section:

 ABSTRACT

Point 1: There is no clarity about the present work, there is mixed information about previous and current work from the same group of authors.

Response 1: References to previously published data are excluded. Now the article presents only previously unpublished information.

INTRODUCTION

Point 1: There is mention of molecular mimicry as mechanism associated to the development of autoimmunity after infection, but they do not mention other mechanisms such as bystander activation.

Response 1: Text added: In addition to molecular mimicry, among the concepts of autoimmunology, other options are considered in the literature as specific mechanisms for the etiology of autoimmunity during the induction of autoimmunity by viruses: epitope spreading [Powell AM, Black MM. Epitope spreading: protection from pathogens, but propagation of autoimmunity? Clinical and experimental dermatology. Vol.26, issue 5, 2001: 427-433], cryptic epitopes [Sercarz EE, Lehman PV, Ametani A. et al. Dominance and crypticity of T cell antigenic determinants. Annu Rev Immunol. 1993;11:729-66.doi: 10.1146/annurev.iy.11.040193.003501], and bystander activation  [Stephan Ehl, Joachim H., Peter A., Hans H., Zinkernagel R. Bystander Activation of Cytotoxic T Cells: Studies on the Mechanism and Evaluation of In Vivo Significance in a Transgenic Mouse Model. J. Exp. Med.  The Rockefeller University Press • Volume 185, Number 7, April 7, 1997: 1241–1251]. Each of these mechanisms is regarded as a likely etiological mechanism responsible for the activation of self-reactive  immune cells. Moreover, a combination of the options mentioned above should not be excluded. In this article, we have focused on the putative  role of molecular mimicry and some pathomorphological correlates.

Point 2: Authors claim that there is a provocative mechanism of pathological autoimmunity triggered by infection with COVID-19, however this is not sustained by the references 14-15, rather reference number 14, reported on peptide sharing between SARS-CoV-2 spike glycoprotein and surfactant-related proteins. And reference 15, reported on 7–9 residue matches in several known human proteins with a 15mer palindromic SARS-CoV-2 peptide. But neither of these references, proves the induction of autoimmunity by the presence of the infectious agent.

Response 2: We have substituted the references mentioned for a reference of newer comprehensive monograph which contains a lot of evidences for autoimmunity provocation by COVID-19 discussing various mechanisms of their relations.

Future of Autoimmunity Research. Autoimmunity,Covid-19, Postcovid19 Syndrome and Covid-19 Vaccination. Volume 1. Eds: Y. Shoenfeld, A. Dotan. Acad. Press: London a.e., 2022: 830 P.

METHODOLOGY

Point 1: Authors used ALIGNMENTAJ, created by M.G.Normatov. However, there is no information about the performance of this tool in other studies. I tried to search for additional articles using this tool and was not able to find any. If authors could present some information about this, it would be reassuring.

Response 1: We created ALIGNMENTAJ specifically for our work, we have not yet published it officially anywhere, so you could not find our program (the software is now pending official registration in RF State Foundation of Algorithms and Programs and will be patented). Nevertheless, for non-commercial scientific purposes we would like to share it with a reviewer. Hence, we showed in the article (in the section “Methodology and materials” an actual  reference giving access to our program, one may found there the details.

Link to ALIGNMENTAJ: https://github.com/muslimb/MyProekt1

Point 2: Immunohistochemical analysis searched for SARS-CoV-2 Spike protein and Caspase 3. However, authors stated that the aim of this manuscript was to explore NP and MP for antigen mimicry with several common target antigens of human autoimmune endocrinopathies (hypophysitis, adrenalitis, insulitis and oophoritis/orchitis). Why are authors including information about the SP, when it was not considered in the objective? Is it because this information has been partially published before (refence #17)?

Response 2: Immunohistochemical analysis was aimed here  just to detect the presence and localization of SARS-CoV-2 in organs responsible for autoimmune endocrinopathies. It is commonly located by means of commercially available antibodies to its main marker antigen which is S-protein. That’s why we checked virus location by its S-protein, not by other proteins, like in almost all other articles by different authors. And the method  revealed the location of virus.Location by M- and N- proteins will not add information as regards to virus presence, being redundant for the purpose of our study. Caspase 3 is checked not for virus location, but for evaluation of cell apoptotic processes.

Point 3: The Immunohistochemical analysis was performed in endocrine organs of patients that died from COVID, although they only confirmed SARS-CoV-2 positivity but no cause of death, patients could have died from different causes and be positive to COVID-19. This is only stated in the results section but should be specified in the methodology section. There is no information about the presence of clinical signs of autoimmunity in these patients either.

Response 3: In all cases COVID-19 was a direct reason of death confirmed by protocol of autopsy (which is obligatory investigation by Russian Federal law in any case of death occurring in a hospital). Post-mortem pathological anatomical examination documented the absence of other pathology incompatible with life. But the mechanism of death in all cases was related to respiratory failure and multi-organ hypoxia. The presence of this was proven by supravital radiological methods (CT) and  confirmed on autopsy as well as the fact that endocrine organs were involved in viral inflammation.

Point 4: The bioinformatic analysis as well as table 5, contain information about the spike protein which have already been published by the same group of authors (reference #17 of the manuscript). This information should be removed as is not in accordance with the objective of the present paper and has already been published.

Response 4: This time new immunoreactive epitopes in the IEDB database for SARS-CoV-2 pentapeptides were revealed, previously not reported.

Point 5: What is the relevance of table 4, 5 and 6. This information is already available in previous tables. The ID in IEDB could just be added to each sequence in previous tables.

Response 5: We added the identification number of immunoreactive epitopes in the previous tables inside square brackets. There will be only three tables in the article, which is a good optimization, thanks to reviewer.

Point 6: Authors state that the absence in the expression of caspase 3 by degenerative and dying endocrinocytes (Fig. 2, I-H), is most likely due to other mechanisms of death, and not apoptosis, but other forms of apoptosis have been demonstrated by decreased levels of Bcl-2 and increased Bax, that directly activates caspase 7 and 9. (Oncogene 20, 6570-6578 (2001) https://doi.org/10.1038/sj.onc.1204815

Response 6: We are agree and have mentioned this fact as well as and the recommended reference:

Of course, the absence of Caspase-3 expression does not entirely exclude the possibility of apoptotic mechanisms of cell death, because in some pathways of apoptosis the process can bypass the Caspase-3 link [Oncogene 20, 6570-6578 (2001) https://doi.org/10.1038/sj.onc.1204815]

Point 7: Figure 2. Letters A, B, and C, D, seem to present the same information. Letters I and F are supposed to have arrows; however, they are not visible. Letters I-H refer to the expression of the spike protein, not included in the objective of the present paper.

Response 7: A, B, C and D are given for more detailed presentation of  the same information. Under letters I and F we have added required arrows now.

DISCUSSION

Point 1: There is still and ongoing discussion about the origins of autoimmunity, and although some authors support molecular mimicry, there are several studies that oppose this mechanism as the origin of autoimmunity and favor others such as bystander activation, or even a combination of both. 

Response 1: Please note that article contains also the opposite point of view. The paper is not a review, that’s why we did not give more details as regards to various mechanisms of viral-induced autoimmunity. But we are of the same point of view on this problem, as the reviewer – and  to make it clear we added appropriate mentions in introduction and in conclusion.

Point 2: Authors from this paper consider molecular mimicry as the sole explanation for the development of autoimmunity but this is far from being generally accepted, mainly because there is still a missing link since more than often mimicry is not enough to ensure an autoimmune response. Apart from rheumatic fever and a few other cases, this mechanism has not been proven to be the origin of other autoimmune diseases, and even less in the case of systemic autoimmune diseases, in experimental models there is always the need to induce an artificial severe inflammatory response for autoimmunity to develop, the single administration of the “autoantigen” almost never is considered enough for the animal to develop autoimmunity, and this should be considered by the authors, otherwise the information contained in the paper might be misleading.  

Response 2: No, authors do not consider mimicry the sole mechanism of autoimmunity and this is stated in several places in the text. But we agree with the reviewer that autoimmunity phenomenon is mosaic and multi – faceted and we added to conclusion the following conclusive statement:

“Autoimmune disorders have a multi-faceted etiology and depend on polygenic predispose and additive action of many epigenetic and environmental factors causing  hyperstimulation of imperfectly functioning immune system. It means that the phenomenon of molecular mimicry cannot be blamed as their single prerequisite, being just a tile in mosaic of autoimmunity.”

Point 3: Authors also claim than an unbalance in the regulatory mechanisms of the immune response triggered by the mimicking can be responsible for immunosuppression to the development of a cytokine storm, nevertheless this is too broad and not just the regulatory mechanisms have been implicated in autoimmunity, but other parts of the immune response including the innate and the adaptive have also been linked to the development of autoimmunity. 

Response 3: We just mean that if the  adrenal cortex functions properly it is a protective factor against hemodynamic shock. If adrenals failed it always aggravate and accelerate  shock development. This is an axiom of battle surgery. Otherwise, why use glucocorticoids for shock treatment and for COVID-related cytokine storm treatment  also?  By the way, glucocorticoid response downregulates the excessive activities of BOTH innate and adaptive immunity. So, we still believe this part of discussion appropriate, because anti-shock potential of proper adrenocortical stress reaction is, to our impression, still underestimated by many clinicians.

Point 4: What seems to be interesting though is that in SARS-CoV-2 infection particularly, is that the infection frequently triggers a very important inflammatory response, and this could certainly lead to a bystander activation, which in conjunction to molecular mimicry could lead to an autoimmune response, this could be added to the discussion.

Response 4: We agree and this is worth to add (which was done).

Point 5: The current criterion for identifying molecular mimicry is from a book published in 2007 (reference #29), to this respect, only the first has been really proved, since the epidemiological link is still debatable.

Response 5: Link replaced:

 Future of Autoimmunity Research. Autoimmunity,Covid-19, Postcovid19 Syndrome and Covid-19 Vaccination. Volume 1. Eds: Y. Shoenfeld, A. Dotan. Acad. Press: London a.e., 2022: 830 P.

Point 6: Although authors mention the importance of structural criteria rather than primary sequence for TCR recognition, there is no account for this in the results of the paper, and this is of upmost importance, since peptides might be hidden or even susceptible to suffer from PTMs and their recognition is not always achieved. 

Response 6: We made necessary additions to text.

Point 7: The presence of the virus and following infiltration of cells might be just a reflection of infection, not really a reflection of autoimmunity, as is being stated by the authors. Therefore, authors must support why they choose the later and not the first as a possible explanation for this finding.

Response 7: As with most other viral infectious diseases, pathological cellular infiltration is not so much a response to cell infection and damage as a manifestation of cell-mediated immune self-damage. At the same time, not only infected cells are always exposed to autoaggression. Moreover, massive infection of cells does not lead to any significant exudative cellular reaction. Based on this, the existing selective (and not massive) pathological cellular infiltration is considered as a manifestation of the implementation of autoimmunity. (see e.g. “Infection and Autoimmunity” by N. Rose and Y.  Shoenfeld, any edition)

The phenomenon of mononuclear infiltration is a consequence of the connection of the immune component. The virus damages the cell. The response to injury is inflammation. Cytokines of inflammation and cytokines of damaged cells may trigger the activation of autoimmunity mechanisms, for example according the “danger model” promoting the expression of co-stimulatory molecules and prolonging the existance of immunosynapses. Also the mechanisms of promotion may include  including epitope spreading, cryptic epitope, and, of course, bystander activation.

Point 8: Authors mention a few “marker autoantigens” associated with autoimmunity but not all where shared with SARS-CoV-2 or any other coronavirus (17-alpha-hydroxylase and CTLA-4).

Response 8: Yes, we objectively give all the results, both positive and negative ones. By the way, quite often excessive autoimmunity just against one autoantigen is enough to precipitate an autoimmune disease. For example, IDDM 1 patients usually do not display full spectrum of autoantibodies against various autoantigens of islets, just having some of them.

Point 9: The lack of expression of proteins from SARS-CoV-2 in the neurohypophysis could be explained by a lack of receptors ACE2 or NRP-1? And if so, what would be the clinical relevance of such distinction in expression?

Response 9: Indeed, despite the detection of mRNA for ACE 2 in the cells of the cerebral cortex, striatum, hypothalamus, and brainstem, there is no evidence of the presence of ACE 2 and NRP1 in the neurohypophysis, but there is also no evidence of their absence there. This, in our opinion, is due to the lack of works devoted to this problem. Without being firmly convinced of the absence of ACE 2 and NRP in the neurohypophysis, it is not yet worth looking for any clinical consequences of the revealed fact from the absence of expression of SARS-CoV-2 proteins in the neurohypophysis. Based on the logic of the question - the absence of prerequisites for infection of a cell (tissue) will make the possibility of their immune damage less likely??

There is a growing number of clinical papers demonstrating that oxytocin is protective in COVID-19 and inproves the outcome [WangS.C.,   Zhang F, Zhu H. et al. Potential of Endogenous Oxytocin in Endocrine Treatment and Prevention of COVID-19. Frontiers in Endocrinology. 2022, vol.13, Article 799521 . doi: 10.3389/fendo.2022.799521 ].

Anyway, product of neurohypophysis oxytocin was successfully used in clinical treatment of COVID-19 (Diep Ph-T., Chaudry M., Dixon A.,Chaudry F., Kasabri V. Oxytocin, the panacea for long-C0VID? A review. Horm Mol Biol Clin Invest 2022; 43(3): 363–371), which may be a kind of clinical relevance of our data.

Perhaps, it can somehow be correlated with the “protection” of neurohypophysis which secrets it from coronavirus invasion.

CONCLUSION

Point 1: The paper lacks a conclusion.

Response 1: Now it has.

Autoimmune disorders have a multi-faceted etiology and depend on polygenic predispose and additive action of many epigenetic and environmental factors causing  hyperstimulation of imperfectly functioning immune system. It means that the phenomenon of molecular mimicry can not be blamed as their single prerequisite, but it  is just a tile in mosaic of autoimmunity.

Round 2

Reviewer 2 Report

I thank the authors for the effort in answering all of my concerns.

I think the manuscript is a bit more cautious about previous assumptions. Unfortunately, I was not able to understand or read some of their references, as they are in Russian or refer to book chapters. The problem with this type of references is that they cannot be accessed easily by international readers, and the support to author's ideas or concepts might be questioned.

For this reason, I would like to invite authors to support their ideas with academic papers in English, whenever possible.

The inserted comments (in red on page 3 and 11) cab be improved, after verifying the grammar and spelling, preferably by an academic English native speaker.

Author Response

Response to Reviewer 2 Comments

Point 1: I think the manuscript is a bit more cautious about previous assumptions. Unfortunately, I was not able to understand or read some of their references, as they are in Russian or refer to book chapters. The problem with this type of references is that they cannot be accessed easily by international readers, and the support to author's ideas or concepts might be questioned.

For this reason, I would like to invite authors to support their ideas with academic papers in English, whenever possible.

 Response 1: On your advice, links to russian journals from the text of the article have been removed and replaced with an international journal (â„– 14) and a book (â„– 1), which are available in google search.

Point 2: The inserted comments (in red on page 3 and 11) can be improved, after verifying the grammar and spelling, preferably by an academic English native speaker.

Response 2: Were improved by an academic English native speaker.

THE AUTHORS ARE SINCERELY GRATEFUL TO THE REVIEWER’S COMMENTS AND ADVICE, WHICH UNDOUBTEDLY MADE THE ARTICLE MUCH MORE ACCESSIBLE TO THE READER.
